# EFFECTIVELY USING PUBLIC DATA IN PRIVACY PRESERVING MACHINE LEARNING

## ABSTRACT

A key challenge towards differentially private machine learning is balancing the trade-off between privacy and utility. A recent line of work has demonstrated that leveraging *public data samples* can enhance the utility of DP-trained models (for the same privacy guarantees). In this work, we show that public data can be used to improve utility in DP models significantly more than shown in recent works. Towards this end, we introduce a modified DP-SGD algorithm that leverages public data during its training process. Our technique uses public data in two complementary ways: (1) it uses generative models trained on public data to produce synthetic data that is effectively embedded in multiple steps of the training pipeline; (2) it uses a new gradient clipping mechanism (required for achieving differential privacy) which changes the *origin* of gradient vectors using information inferred from available public and generated data from generative models. Our experimental results demonstrate the effectiveness of our approach in improving the state-of-the-art in differentially private machine learning across multiple datasets, network architectures, and application domains. Notably, we achieve a $75.1\%$ accuracy on CIFAR10 when using only $2,000$ public images; this is *significantly higher* than the state-of-the-art which is $68.1\%$ for DP-SGD with the privacy budget of $\varepsilon = 2, \delta = 10^{-5}$ (given the same number of public data points).

## 1 INTRODUCTION

Machine learning is becoming an essential part of many technological advancements in various fields. One major concern with usage of machine learning is privacy of individuals whose data is used to develop the machine learning models. To tackle this concern, recent works De et al. (2022); Kurakin et al. (2022); Abadi et al. (2016); Yu et al. (2021); Amid et al. (2021); Li et al. (2022a) suggest to train ML models with *differential privacy* (Dwork et al. (2014)) guarantees. However, existing differential privacy (DP) techniques for ML impose large degradation to the utility of the trained models in comparison to non-private models. Recent works have made substantial improvements to the utility-privacy trade-off of such private ML techniques, (e.g., by scaling the hyper-parameters (De et al. (2022))), however, there still exists a huge gap between the accuracy of DP-guaranteeing ML mechanisms and their non-private alternatives, e.g., De et al. (2022) achieves an accuracy of $65\%$ on CIFAR10 (for $\varepsilon = 2.0$ and $\delta = 10^{-5}$) compared to the $> 90\%$ accuracy of non-private models.

In this work, we explore an emerging approach to close the utility gap between private and non-private models. Specifically, recent works (De et al. (2022); Kurakin et al. (2022); Abadi et al. (2016); Yu et al. (2021)) show that leveraging *publicly available (therefore, non-private) data* can enhance the utility of DP-trained models without impacting their privacy guarantees. In such works, the public data is used to *pre-train* the model, and then the pre-trained model is fine-tuned with the private data while applying DP protections.

In this work, we show that, while recent works use public data to pre-train private models, public data can be used much more effectively in enhancing the utility of private models. To this aim, we design a generic method to utilize public data in differentially private machine learning, an approach we call *Differentially Private Origin Estimation Stochastic Gradient Descent (DOPE-SGD)*. Our work uses two complementary techniques to enhance the utility of differentially private models. First, it improves the quality of the noisy gradients based on the available non-private data. This helps by reducing the variance of the noise added to the gradients in the DP model, therefore better preserving

the information in the original gradient vector (Section 3). Second, DOPE-SGD uses advanced data augmentation techniques to enhance the quality of the data used for training, therefore reducing overfitting to the public data and improving generalization.

Through extensive experiments we show that DOPE-SGD's use of public data along with data augmentation improves the privacy-utility trade-offs of private models by large margins. For instance, we show improvements up to $12.3\%$ over DP-SGD models on the CIFAR10 dataset, pre-trained with the same public data. We also show improvements on language model both on training from scratch (from 221 to 198 on a small BERT model) or fine-tuning (from 21.23 to 19.09 on GPT-2) with $\varepsilon = 1.0$ and $\delta = 10^{-5}$.

## 2 BACKGROUND

Differential privacy (Dwork (2011); Dwork et al. (2014)) is the gold standard for data privacy. It is formally defined as below:

**Definition 1** (Differential Privacy). *A randomized mechanism $\mathcal{M}$ with domain $\mathcal{D}$ and range $\mathcal{R}$ preserves $(\varepsilon, \delta)-$differential privacy iff for any two neighboring datasets $D, D' \in \mathcal{D}$ and for any subset $S \subseteq \mathcal{R}$ we have:*

$$\Pr[\mathcal{M}(D) \in S] \le e^{\varepsilon} \Pr[\mathcal{M}(D') \in S] + \delta \tag{1}$$

*where $\varepsilon$ is the* privacy budget *and $\delta$ is the* failure probability.

Several works have used differential privacy in traditional machine learning algorithms to protect the privacy of the training data (Li et al. (2014); Chaudhuri et al. (2011); Feldman et al. (2018); Zhang et al. (2016); Bassily et al. (2014)). Many of these works (Feldman et al. (2018); Bassily et al. (2014); Chaudhuri et al. (2011)) use properties such as convexity or smoothness for their privacy analysis, which is not necessarily true in deep learning, and therefore, one cannot use many of such methods in practice. Abadi et al. (2016) designed a deep learning training algorithm, DP-SGD, where they used gradient clipping to limit the sensitivity of the learning algorithm, and then add noise to a clipped model gradient proportional to its sensitivity. As we know, training a deep learning model is an iterative process. The main approach to analyze the privacy cost of private deep learning is to compute the privacy cost of the single step of the learning algorithm and then use composition method to calculate the overall privacy cost which is commonly done in RDP ( Mironov (2017)) instead of $(\varepsilon, \delta)$-DP. One of the important features of differential privacy is the post-processing ( Mironov (2017); Dwork et al. (2014)) which we will utilize in this work. DP-SGD is now commonly used to train differentially private deep learning models.

## 3 DOPE-SGD: OUR IMPROVED DPSGD ALTERNATIVE

We can improve the utility-privacy trade-off for differentially private machine learning algorithms in three phases: (1) use *pretraining* to improve the initial point of private training, (2) use *better algorithms* for differentially private training, (3) do a *post processing* on the private models. Previous works showed the effect of using pretraining (Abadi et al. (2016); Kurakin et al. (2022); De et al. (2022)), so in this work we mainly focus on two last phases of private training.

### 3.1 DP-SGD WITH ADAPTIVE ORIGIN

One of the main steps of DP-SGD is to clip the gradient of each instance and add noise to the gradient of each batch based on a clipping value. Clipping the gradient results in a bias in the optimization process which makes convergence slower. One way to prevent this is to use larger clipping values, however, this requires larger noise to obtain the desired privacy. The main idea of this work is to *clip the gradients around an estimate of the gradient* instead of clipping the gradient around the origin, as shown in Figure 1. As a result, we can potentially clip more harshly (i.e., use small clipping values) around the carefully chosen centers with less bias in the optimization compared to DP-SGD and obtain better accuracy while maintaining privacy.

We first introduce a general algorithm called DP-SGDA (Algorithm 1) that uses adaptive origin selection. We use a function $\mathcal{G}$ that takes the history of the protocol and also some auxiliary

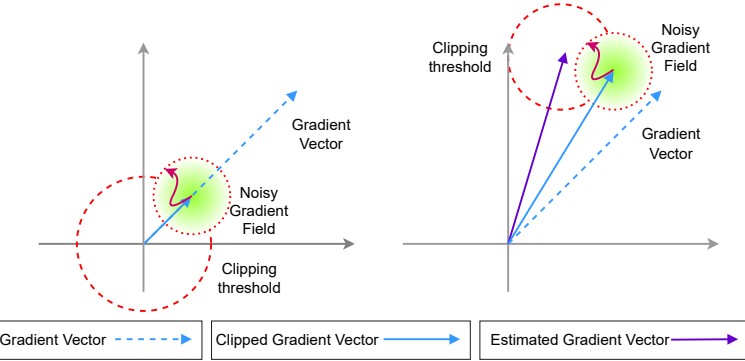

Figure 1: Clipping the gradients around an estimation of the origin (purple vector) can increase the ratio of the clipped gradient vector to the added noise. As a result the final privatized gradient vector is closer to original gradient vector compared to clipping around zero.

---

**Algorithm 1** DP-SGD with Adaptive Origin (DP-SGDA)

---

**Require:** training dataset $D$, adaptive origin function $\mathcal{G}$, batch size $n$, learning rate $\eta$, noise scale $\sigma$, gradient norm clip $C$, loss function $l$, auxiliary information $aux$

1: Initiate $\theta$ randomly
2: **for** $t \in \{T\}$ **do**
3:   $B_t \leftarrow$ sample $n$ instances from dataset $D$
4:   $\nabla[t] \leftarrow \vec{0}$
5:   $\hat{g} \leftarrow \mathcal{G}(\nabla[1], \ldots, \nabla[t-1], aux)$.
6:   **for all** $(x, y) \in B_t$ **do**
7:    $\nabla^{(x,y)} \leftarrow$ gradient of $l(x, y)$
8:    $\overline{\nabla^{(x,y)}} \leftarrow \frac{(\nabla^{(x,y)} - \hat{g}) \times C}{\max(C, \|\nabla^{(x,y)} - \hat{g}\|_2)}$
9:    $\widetilde{\nabla^{(x,y)}} \leftarrow \hat{g} + \overline{\nabla^{(x,y)}}$
10:    $\nabla[t] \leftarrow \nabla[t] + \widetilde{\nabla^{(x,y)}}$
11:   **end for**
12:   $\nabla[t] \leftarrow \nabla[t] + \mathcal{N}(0, \sigma^2 C^2 \mathbb{I})$
13:   $\theta \leftarrow \theta - \eta \widetilde{\nabla[t]}$.
14: **end for**
15: **return** output $\theta$

---

information as input and outputs a point $\hat{g}$ that will be used as the origin for the clipping operation. The following proposition states that any instantiation of this algorithm with this algorithm will satisfy DP guarantees.

**Proposition 2.** *For any function $\mathcal{G}$, DP-SGDA (Algorithm 1) obtains the same DP and RDP guarantees as DP-SGD (Abadi et al. (2016)), when the clipping threshold $C$, sub-sampling rate $q$ and noise parameter $\sigma$ are equal (See Appendix F for detailed proof).*

**Theoretical Justification**  To show the benefit of our Algorithm 1, we consider a setting of the lipschitz loss function with concentrated gradients.

**Definition 3** (Lipschitz and gradient-concentrated loss function). *A loss function $\ell$ defined on a model space $\Theta$ and input space $X$ is $\mathcal{L}$-lipschitz if for all $x \in X$ and $\theta \in \Theta$, $\|\frac{\partial \ell}{\partial \theta}(\theta, x)\|_2 \leq \mathcal{L}$. The loss function is $r$-concentrated if for all $\theta \in \Theta$, there exists a point $c_\theta$ in the gradient space such that we have*

$$\forall x \in X; \|\frac{\partial \ell}{\partial \theta}(\theta, x) - c_\theta\|_2 \leq r.$$

*Note that any $\mathcal{L}$-lipschitz loss function is $\mathcal{L}$-concentrated. We call $c_\theta$ the concentration point for $\theta$. We call an oracle function $C(\cdot)$ a concentration point oracle if given a model $\theta$, it returns $C(\theta) = c_\theta$.*

---

**Algorithm 2** Differentially Private Origin Estimation-SGD (DOPE-SGD)

---

**Require:** private training dataset $D$, non-private data $D_s$, non-private batch size $n_s$, learning rate $\eta$, private batch size $n$, noise scale $\sigma$, gradient norm clip $C$, loss function $l$

1: Initiate $\theta$ randomly
2: **for** $t \in \{T\}$ **do**
3:     $B_t \leftarrow$ sample $n$ instances from dataset $D$
4:     $B_s \leftarrow$ sample $n_s$ instances from dataset $D_s$
5:     $\nabla_\theta^s \leftarrow \nabla L(B_s)$
6:     $\nabla_\theta^G \leftarrow \vec{0}$
7:     **for all** $(x, y) \in B_t$ **do**
8:         $\nabla_\theta^{(x,y)} \leftarrow$ gradient of $l(x, y)$
9:         $\widetilde{\nabla_\theta^{(x,y)}} \leftarrow \nabla_\theta^s + ((\nabla_\theta^{(x,y)} - \nabla_\theta^s) \times C / \max(C, \|\nabla_\theta^{(x,y)} - \nabla_\theta^s\|_2))$
10:         $\nabla_\theta^G \leftarrow \nabla_\theta^G + \widetilde{\nabla_\theta^{(x,y)}}$
11:     **end for**
12:     $\widetilde{\nabla_\theta^G} \leftarrow \nabla_\theta^G + \mathcal{N}(0, \sigma^2 C^2 \mathbb{I})$
13:     $\theta \leftarrow \theta - \eta \widetilde{\nabla_\theta^G}$
14: **end for**
15: **return** output $\theta$

---

Now we state the following proposition about the privacy cost of our algorithm in comparison with the privacy cost of DP-SGD.

**Proposition 4.** *Let $X$ and $\Theta$ be example and model spaces and let $\ell$ be a $\mathcal{L}$-lipschitz and $r$-concentrated loss function for $X$ and $\Theta$. Optimizing this loss function with clipping threshold $r$ in Algorithm 1 with concentration point oracle used as the adaptive origin function, will induce the same output (model) distribution as training with DP-SGD with clipping threshold $\mathcal{L}$, as long as all examples in the training set are in $X$. Algorithm 1 achieves $(c\frac{rq}{\sigma}\sqrt{T\ln(1/\delta)\ln(T/\delta)}), \delta)$-DP, where as DP-SGD achieves $(c\frac{\mathcal{L}q}{\sigma}\sqrt{T\ln(1/\delta)\ln(T/\delta)}), \delta)$-DP, for a constant $c$, sampling rate $q$, and number of iterations $T$ and sufficiently large $\sigma$ (See Appendix F for detailed proof).*

Proposition 4 shows that our algorithm can reduce the sensitivity of the gradient update at each iteration from $\mathcal{L}$ to $r$. Given that $r < \mathcal{L}$, our algorithm would obtain better privacy guarantees when the gradients are concentrated. Although these bounds are stated based on advanced composition theorem for approximate differential privacy, the privacy benefit of our algorithm holds for alternative notions of privacy, such as RDP, because of the reduced sensitivity in our algorithm.

**Instantiation with (augmented) public data.** To obtain the estimate of gradient vectors, we use the idea of *(augmented) public data*. Public data has been used previously in many traditional differentially private tasks. In this work, we show using public data to estimate can improve deep learning with differential privacy significantly. Algorithm 2 summarizes the design of our approach. We also take advantage of advanced augmentation techniques (such as diffusion models) to utilize the public data further. The exact data augmentation mechanism heavily depends on the specific task and type of the dataset. We can use several techniques for data augmentation, which can range from using a basic shifting of images to designing a synthesizer (Mood (1950)) using complex generative models (Nichol & Dhariwal (2021)).

## 3.2 ENSEMBLE OF PRIVATE MODELS

One of the properties of the way we analyze DP-SGD at the current state is that we assume every step of the iterative training is public and an adversary can use all of the intermediate steps for an attack. This might not be true in usual training of a deep learning model, however, this assumption is currently necessary for the analysis. Although this assumption might increase the privacy budget (Nasr et al. (2021)), it will allow us to use all of the models in all iterations of the training without additional privacy cost (due to post-processing feature of differential privacy). One example of taking advantage of this fact is using "Exponential Momentum Averages (EMA)" approach in differential private

models (De et al. (2022)). We also show that by ensembling the intermediate models we can further improve the accuracy of the predictions. We use two approaches to ensemble the models. First, we use the idea of majority voting in which to give a label for each input we query the last $n$ models and we use the label with the highest number of the votes for classification tasks (in the case that we are interested in the value of logits, such as language models, we take the average of logits among the last $n$ models). Second, we show the effectiveness of taking the average models of last $n$ models and evaluating the inputs on the average instead of the target model.

While the main objective of our work is on the best ways to utilize the public data in the training private models, this approach can be also used in cases where we do not have any additional data. We explore this in Appendix E. In the main text we always use the maximum between the two ensemble methods, but as shown in Appendix E we see that both methods have very similar performance and using majority voting achieves slightly higher performance.

**Why do ensemble methods work?** Stochastic gradient descent with averaging has been studied as a way to cope with the excessive variance coming from stochasticity of SGD Bach (2014). One would expect the reduction of variance by averaging the resulting parameters of the last few models, assuming that the variance is caused by independent noise. Bach (2014) shows faster convergence on convex models by doing the simple averaging operation. In DP-SGD and its variants, a large portion of variance in the model comes from the added noise, which is independently sampled and added to the model after each iteration. This makes DP-SGD suitable for averaging methods as the variance is mostly the result of independent noise. It is important to note that in privacy analysis of DP-SGD and its variants, we always "pay" the privacy cost of all the intermediate models and using them as an ensemble will not violate the privacy guarantees.

We also note that several recent studies have tried to understand the privacy benefits of having hidden states during the course of training and not leaking the intermediate models Feldman et al. (2018); Ye & Shokri (2022). They conjecture that a portion of privacy cost spent for publishing the intermediate models in DP-SGD might be wasted, leading to sub-optimal trade-off between privacy and accuracy. The success of our ensemble methods can be seen as a verification of this conjecture and shows the possibility of better ways to use the privacy budget.

## 4 Experiments

To evaluate the suggested techniques we consider two main scenarios, where public data is coming from a similar distribution as the training dataset and also where the data is from a slightly different distribution. In the main body of this work, we focus on the CIFAR and WikiText datasets as they represent a majority of the applications of the deep learning models. In Appendix B, we show the effectiveness of our approach for other architectures and tasks.

Additionally, we show that we can further improve the utility of the private models by including the public data in the training dataset. This way, with a fixed batch size, we can use a smaller sub-sampling rate in our privacy analysis. Please note that we also do pretraining on the public dataset. However, we found that while many of the public instances have zero loss after the pretraining it is still useful to include the public dataset in the training dataset. We explore this in our experiments in detail.

**CIFAR10 dataset.** De et al. (2022) showed that using a WideResNet architecture for the CIFAR10 dataset can improve the state-of-the-art differentially private model significantly. Therefore, we also use similar architecture for the majority of our experiments. In Appendix B, we also evaluate a smaller convolution neural network. Due to the computation limitation we use WideResNet16-4 as explained in De et al. (2022) since there gain of using the larger model is not very significant.[1] We also use CIFAR100 as out-of-distribution public data.

**WikiText-2 dataset.** Our dataset setup mainly follows the previous work (Amid et al. (2021)). The texts are tokenized by the top 8k most frequent words and a special token for the remaining words. The dataset is then constructed as length-35 sequences and we have 59,675 data points in the

---

[1]Training WRN40-4 on eight A100 in our setting takes more than 96 hours.

Table 1: An ablation study on the effect of different techniques using training data for models trained on CIFAR-10 under $(2, 10^{-5})$-DP. We use WRN16-4 with all of the augmentation and optimization techniques as detailed in De et al. (2022). We use 2,000 images of the CIFAR10 training dataset ($4\%$ of the whole dataset) as the public data which is sampled uniformly from each class. The public dataset is augmented to 40,000 instances.

| Setting | Test Acc (%) |
| --- | --- |
| Baseline (WRN16-4 De et al. (2022)) (cold) | 64.9 |
| + Pretraining on the public data (warm) | 68.1 |
| + Pretraining on the generated data using public data (warm-aug) | 72.0 |
| + Including the generated data in the training dataset (extended) | 73.7 |
| + Using DOPE-SGD (Algorithm 2) | 74.8 |
| + Using Ensemble models | 75.1 |

training set. We use 4% of the training set as in-distribution public data and WikiText-3 as imperfect out-of-distribution public data (we removed the overlapping part of WikiText-2 from WikiText-3). We use a BERT-based Devlin et al. (2019) model with two blocks.

**Settings:** We show the results in several settings. In our experiments we use the term "*warm*" to describe a setting where we first train a model without any privacy on the non-private data (as opposed to "*cold*" where we do not pre-train). We also use "*extended*" to note a case where we augment the training dataset by including the non-private dataset (which we use the augmented data). Please note that we did hyper-parameter tuning for each setting (as detailed in Appendix A). In the main body of this paper we focus on in-distribution results for CIFAR10 and WikiText-2 datasets. We evaluate the other datasets and out-of-distribution datasets in Appendices B and D.

## 4.1 RESULTS

Using the in-distribution public dataset can improve the utility of differentially private machine learning significantly. However, one of the downsides of requiring an in-distribution public dataset is the cost of acquiring such data. As a result, assuming that we can have access to large in-distribution public data is very strong. Therefore, we only assume that we have access to very limited in-distribution public data in this section. However, in order to address the issue of small public dataset and prevent overfitting models on the public dataset, we use various augmentation techniques on the public dataset. In this section, we use a small portion of the target's training dataset as the public dataset.

**Main in-distribution Results.** In addition to traditional augmentation techniques , we train a generative model on the public dataset which can be seen as an ideal augmentation technique. This way, at each iteration, we can generate fresh samples from the generative model and calculate the average gradient over those samples. In particular for image datasets we used DDPM generative models Nichol & Dhariwal (2021), DDPM models are currently slow in generating examples, therefore, in our implementations we generate a large set of data point from the generative models and use a random subsample of that for training the target model.

To understand the effect of each method described in this work, first we do ablation study when using different methods and algorithms in Table 1 for CIFAR10 dataset. One of the main approaches of using public data is to pretrain the target model first on the public data which does not have any privacy cost and then do the private training. Since the public data for the in-distribution dataset is limited (while we are still using traditional augmentation methods) there is only a $3.5\%$ increase in the accuracy. However, we apply our idea of using generative models to augment the public data we see that we can further improve. One important step that to the best of our knowledge has not been used before is to combine the public dataset (or the generated dataset on public data) with the private data that is used during the private training. As shown in Table 1 only by including the public data in the training dataset we can increase the accuracy ($1.7\%$).

When we combine all of the methods for CIFAR10 we increase the accuracy of classification by 7.6 points when using only $4\%$ of the data compared to the case when we pretrain on the training dataset

Table 2: Test accuracy/perplexity for models trained with differential privacy with $\delta = 10^{-5}$ and using $4\%$ of training of CIFAR10/WikiText-2 dataset as the public data. The accuracy of the model trained on only augmented CIFAR10 dataset is $69.4\%$ and perplexity is 240 on Wikitext-2 dataset.

| | CIFAR10 (Test Acc) | | | | WikiText-2 (Test Ppl) | | |
| --- | --- | --- | --- | --- | --- | --- | --- |
| $\varepsilon$ | DPSGD (cold) | DPSGD (warm) | DPSGD (warm-aug) | DOPE-SGD | DPSGD (cold) | DPSGD (warm) | DOPE-SGD |
| 1.0 | 56.8% | 60.1% | 70.0% | **72.1**% | 240 | 221 | **198** |
| 2.0 | 64.9% | 68.1% | 72.0% | **75.1**% | 220 | 206 | **184** |
| 4.0 | 71.9% | 72.4% | 76.0% | **77.9**% | 210 | 183 | **177** |
| 6.0 | 77.0% | 77.1% | 78.7% | **80.0**% | 190 | 167 | **156** |

Table 3: Test accuracy of models trained with differential privacy with $\delta = 10^{-5}$ and using $4\%$ of training of CIFAR10 dataset as the public data for different training algorithms.

| | | Original | Ours | |
| --- | --- | --- | --- | --- |
| $\varepsilon$ | Method | Setting | Augmentation | DOPE-SGD |
| $\varepsilon = 2$ | Mirror Gradient Descent Amid et al. (2021) | 68.7% | 70.5% | 75.1% |
| | Gradient Scaling Li et al. (2022a) | 68.7% | 69.1% | |
| $\varepsilon = 4$ | Mirror Gradient Descent Amid et al. (2021) | 73.1% | 74.5% | 77.9% |
| | Gradient Scaling Li et al. (2022a) | 73.5% | 74.1% | |
| $\varepsilon = 6$ | Mirror Gradient Descent Amid et al. (2021) | 77.2% | 78.2% | 80.0% |
| | Gradient Scaling Li et al. (2022a) | 77.9% | 78.1% | |

which is the common practice. We also evaluated our results for different privacy budgets as shown in Table 2. Our results suggest that the common practice of just using the public dataset to pretrain the models is an ineffective way of using the public data and by properly using this additional information we can improve the state of the private training significantly and we can achieve practical accuracies even at more private regions. When we increase the privacy budget, the gap between our approach and DP-SGD reduces which is excepted. Because in the higher privacy regimes we are adding less noise, therefore, the model has an easier time learning from the private data and does not rely on the public data as much. As we also show in Appendix B our methods are also effective in smaller models, different datasets and imperfect datasets. One of the major downsides of our method is the increase in the computation cost which we will discuss in Appendix C. In Appendix D, we also show that when the public data is not from the same distribution, we do not see the same improvement as in-distribution data and we introduce changes in our approaches to improve the results.

**Comparison.** Recent works also proposed better algorithms to take advantage of public dataset. Amid et al. (2021) uses the idea of mirror gradient descent in traditional machine learning and extends the idea for deep models. All of the recent work and our work use public data to estimate the geometry of the gradient field. All of the recent work used a different approach to approximate the geometry estimation using the public data. Amid et al. (2021) used the gradient of public data for first order approximation of the gradient geometry. Li et al. (2022a) used the public data to re-scale the gradients of private data. In contrast our approach uses the gradient of the public data to recenter the private gradients. We compare the accuracy of different approaches on the CIFAR10 dataset. Unfortunately, previous works all use different architecture to evaluate their method and none of them used the current state-of-the-art architecture, therefore, we can not directly compare the accuracy numbers. However, we did our best to train and do a hyper-parameter tuning of both of the previous works for a fair comparison. We used the same public dataset in all of the training runs.

Table 3 summarizes the results of our comparison. Original Setting refers to a setting where we use the method from the previous works and replace the neural network with the current state-of-the-art method and do hyper-parameter tuning. As shown in Table 3 since the previous works do not use our data augmentation techniques there is a huge gap between our approach and previous works. Moreover, we showed that our augmentation techniques can also be used in any public data training approach and can improve the accuracy of the existing mechanism as well. Finally, we see that using all of the techniques our approach achieves higher accuracy compared to the existing methods. In our

Table 4: Test accuracy of models trained with differential privacy using different amounts of public data that is augmented to 40,000 instances on CIFAR10.

| Public Data Size | $\varepsilon = 2$ | $\varepsilon = 4$ | $\varepsilon = 6$ |
|---|---|---|---|
| 500 (1% ) | 68.9% | 72.1% | 77.1% |
| 1,000 (2%) | 73.5% | 74.9% | 78.4% |
| 2,000 (4%) | 75.1% | 77.9% | 80.0% |

Table 5: Test accuracy of models trained with differential privacy using different amounts of augmented data from $2,000(4\%)$ public data on CIFAR10.

| Augmented Data Size | $\varepsilon = 2$ | $\varepsilon = 4$ | $\varepsilon = 6$ |
|---|---|---|---|
| 5K | 69.5% | 73.8% | 78.4% |
| 10K | 70.1% | 75.3% | 79.3% |
| 20K | 70.9% | 77.4% | 79.9% |
| 40K | 75.1% | 77.9% | 80.0% |

Table 6: Finetuning WikiText-2 on pretrained large language models using $4\%$ of the training dataset as public data.

| | WikiText-2 | |
|---|---|---|
| $\varepsilon$ | DPSGD (warm) | Ours |
| 0.25 | 21.30 | 19.16 |
| 0.5 | 21.29 | 19.14 |
| 1.0 | 21.23 | 19.09 |
| $\infty$ | 15.40 | |

Table 7: Test accuracy models trained with user-level differential privacy with $\delta = 10^{-5}$ on EMNIST using $2\%$ users as public users.

| $\varepsilon$ | DP-SGD (all) | DP-SGD (only-private) | DOPE-SGD |
|---|---|---|---|
| 0.5 | 82.2% | 83.1% | 84.7% |
| 1.0 | 83.4% | 84.6% | 85.2% |
| 2.0 | 85.0% | 85.5% | 86.0% |
| 4.0 | 86.2% | 86.4% | 87.1% |

comparison using mirror gradient descent approaches achieves very high accuracy as well when we add all of our techniques on top. Surprisingly Gradient Scaling approach does not scale as well as Mirror gradient descent when we add our techniques. One issue might be that the Gradient Scaling approach is not optimized for the cases where the size of public data is large.

**Effect of Size of the Public Dataset.** The size of the public dataset can affect the utility of learning significantly. We studied the effect of different amounts of public data in Table 4. As expected by having a larger public dataset, the model can achieve better utility.

**Effect of Size of Generated Data.** One of the interesting trade-offs is to see how much the utility of a private model changes by increasing the number of augmentation that we generate for a given public dataset. Table 5 shows the results of this study and as we can see adding more augmentations of the public data in the training dataset increases the utility of the training. However, one of the main limiting factors of using more augmentations is the computation cost of the augmentation. Specially, the current DDPM models are costly in both training and generating different augmentations.

**Pretrained Large Language Models.** All of the state-of-the-art models in language tasks currently use a pretrained large language model and fine-tune them on their specific tasks. While this means that they are using a much larger training dataset, it is still very important to show we can see the same improvements as in previous experiments in this setting. To this aim, we used a pretrained GPT-2 Radford et al. (2019) model (which is not trained on WikiText) and fine-tuned it using both DP-SGD and DOPE-SGD using $4\%$ public data and when we also first train it on the public data without any privacy cost and then use private training. Table 6 summarizes our results, as we can see DOPE-SGD also outperforms DP-SGD in this setting even at small epsilons.

**Extending to Federated Learning with User-Level Privacy.** Another example of settings where we can use non-private data is in federated learning. In many federated applications, users can indicate that they do not require any privacy protections, while this is only a small number of the users, the learning algorithms can benefit significantly from such users. Amid et al. (2021) suggested using such users to first pretrain the model, however, this is not a practical approach since the data of these users will change during the training of a federated learning task. Pretraining the model only on the non-private users will slow down the federated learning process and it won't be as effective. Therefore, our evaluations of federated learning settings focuses on the cases where we do not pretrain on the data (cold setting).

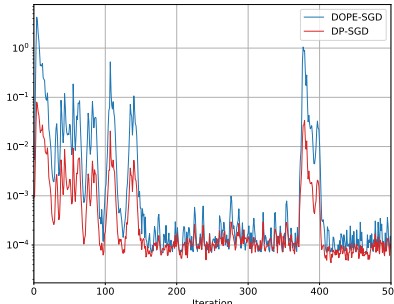

Figure 2: Comparison of DOPE-SGD and non-private's dot product with DP-SGD and non-private's dot product. Using DOPE-SGD will reduce the effect of the noise added by the Gaussian mechanism which can lead to a better performance (all experiments are in warm-aug setting).

For this setting, we used the EMNIST dataset which has 70,000 users and we uniformly divide the instances between the users and we assume a fixed 2% of the users are non-private. We use a WideResNet model for this task and Fed-SGD McMahan et al. (2017) to train this model. In each round of the training we select a subset of users to update the model. We assume the computations and communications are done in a trusted environment, therefore, we can use the *user-level* central differential privacy definition to analyze this setting.

Table 7 summarizes the results and the comparison with the existing approaches that can be applied in the federated learning setting. We compare our approach to using DP-SGD for this setting and show that by leveraging the public data we can achieve better results. One alternative to DP-SGD is to not apply differential privacy on the public user and only apply it on the private users. We compared to this baseline and we showed better performance. This result also shows the benefit of the Algorithm 2 even when we do not use the other techniques described in this work.

**Why DOPE-SGD Works?**    To answer this question we compute the inner product of the gradients of each batch when using DOPE-SGD with non-private gradients of the same batch and compare it with inner product of the DP-SGD and non-private gradients (average between 5 independent runs and after pretraining on public dataset). Figure 2 shows this comparison, as we see the dot product of DOPE-SGD is larger compared to DP-SGD especially in earlier iterations. As a result, we can achieve better convergence rates and privacy-utility trade-off.

## 5    CONCLUSION

As many of previous works showed that deep learning models leak information about their training datasets. New regulations will require the models to protect the privacy of the users that is being used to train large deep learning models. As a result researchers are using differential privacy to train large models. However, the main limiting factor of using differentially private machine learning is the degradation of the utility in most cases. To improve the utility-privacy trade-off, recent works started to use non-private data. The common practice of using public data is currently to just use it to pre-train the model and then use private learning on a private dataset. In this work, our main goal is to show this approach is not the optimal technique to take advantage of non-private data in many cases. We can summarize our work in three practical steps that can be applied in any setting that uses non-private data. First, we should **use advanced augmentation techniques** (which include data synthesis approaches such as GAN, DDPM) on the public data to synthesize more non-private data points. Second, we should consider **including the non-private dataset in the private data during** the private training and finally, by **using more advance private learning algorithms** (e.g, DOPE-SGD, Amid et al. (2021), Li et al. (2022a) ) which are designed to leverage public data to achieve better trade-offs. Using these methods we showed that we can improve the state-of-the-art in private learning significantly.

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

## A  IMPLEMENTATION DETAILS

We implemented Algorithm 2 and the related works in JAX (Bradbury et al. (2018)) and we implemented Algorithm 2 in Opacus (Yousefpour et al. (2021) and private-transformers library Li et al. (2022b)).[2] Experiments in this work are the average between 5 independent runs. For each setting we did a grid hyper-parameters search as mentioned in Table 8 and picked one with the highest average. While doing hyper-parameter training will increase the privacy cost, we didn't consider this in our privacy calculation similar to the previous works (Abadi et al. (2016); Amid et al. (2021); De et al. (2022); Kurakin et al. (2022)).

Table 8: Set of hyper-parameters used in the hyper-tuning phase.

| Parameter | Values |
|---|---|
| Learning rate | [1,2,3,4,5,5.5,6] |
| Noise multiplier | [1,2,3,4,5,8] |
| Public data sample size | [80,160,640,1280,2560] |
| Clipping norm | [0.5,0.8,1.0,1.5] |
| Batch size | [512,1024,2048,4096] |

We summarize different settings in the paper in the followings:

**Cold:** This refers to a setting where we do not pre-train on the available non-private dataset.

---

[2]The code is available at (will be added after publishing).

Table 9: Comparison of the results on CIFAR10 dataset in different settings for both DP-SGD and DOPE-SGD using ConvNet architecture (* the setting that does not utilize the public data). Public data in these experiments are 2,000 images from the dataset which are augmented to 40,000 images.

| $\varepsilon$ | DP-SGD (Abadi et al. (2016)) | | | | DOPE-SGD (Alg.2) | | | |
|---|---|---|---|---|---|---|---|---|
| | cold* | warm | cold-ext. | warm-ext. | cold | warm | cold-ext. | warm-ext. |
| 1.0 | 46.7% | 61.4% | 48.0% | 66.6% | 67.2% | 62.7% | 67.8% | **70.3**% |
| 2.0 | 50.1% | 64.2% | 51.9% | 68.0% | 70.1% | 70.4% | 71.8% | **73.0**% |
| 4.0 | 54.7% | 66.4% | 57.4% | 68.1% | 75.7% | 74.6% | **77.0**% | 73.2% |
| 6.0 | 58.4% | 68.4% | 59.4% | 68.7% | 76.2% | 75.1% | **78.1**% | 76.2% |

Table 10: Comparison of the results on Purchase10 dataset in different settings for both DP-SGD and DOPE-SGD using Fully-Connected architecture. Public data in these experiments are 10,000 records.

| $\varepsilon$ | DP-SGD (Abadi et al. (2016)) | | | | DOPE-SGD (Alg.2) | | | |
|---|---|---|---|---|---|---|---|---|
| | cold* | warm | cold-ext. | warm-ext. | cold | warm | cold-ext. | warm-ext. |
| 1.0 | 30.2% | 32.6% | 34.3% | 36.8% | 33.2% | 35.3% | 40.7% | **42.1**% |
| 2.0 | 34.1% | 35.7% | 37.9% | 39.2% | 35.1% | 37.4% | 44.6% | **45.6**% |
| 4.0 | 44.4% | 45.9% | 47.7% | 49.2% | 45.2% | 48.5% | 50.2% | **52.3**% |

**Warm:** We call a setting warm where we first the model on all available non-private data (both augmented and original public data) and then train only on the private dataset.

**Warm-aug:** Similar to the previous setting we first train the model on the non-private dataset and then train it on both non-private and private dataset.

**Extended:** We refer to a setting where we include the non-private (which includes the augmented dataset) dataset in the private training dataset.

## B EVALUATING DIFFERENT ARCHITECTURES AND DATASETS.

To show that our techniques can also be applied to a simpler model, we used a ConvNet network with three layers on CIFAR10 dataset with batch size of 256. Table 9 presents the accuracy of this model in different settings as used in Table 2. Similar to the results in Table 2 we see impressive improvement over DP-SGD.

We showed the effectiveness of our approach to the popular vision and language modeling tasks. However, many practical applications use categorical data. To show the effectiveness of our approach in such tasks, we focused on the Purchase dataset which is the shopping records for several thousand individuals. Each data record corresponds to one customer and has 600 binary features, we use Purchase10 which has 10 classes. Similar to previous works Shokri et al. (2017); Nasr et al. (2018), we use a 4 layers fully connected network. We used 90,000 instances as the training dataset, 5,000 instances as the public data and 100,000 instances as training dataset. Table 10 summarizes the results of our experiments. As we can see, the overall accuracy of this task is much lower than the previous datasets. However, we still see a significant gain when using public-data and using Algorithm 2.

## C COMPUTATION COMPLEXITY

The main downside of our approach is the additional computation cost compared to the DP-SGD. There are two main components to these additional costs. First, the costs of the augmentation. As we know, augmentation techniques are used widely in training deep learning models. In this work we try on using diffusion models as our augmentation technique which have high computation costs. This can be the main source of the additional computation cost for our work. As a reference, training a diffusion model Nichol & Dhariwal (2021) on 2,000 cifar10 on a single V100 takes about 48 hours. Also generating 40,000 images using this model takes about a week of compute. The second

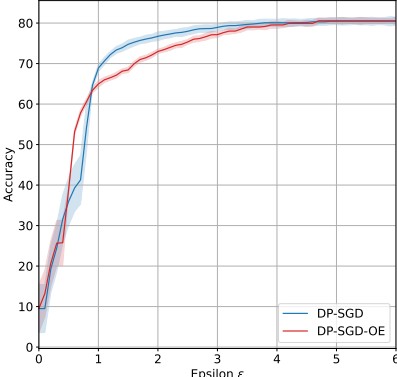

Figure 3: The effect of out-of-distribution data on Algorithm 2 without clipping the public gradient (Equation 2).

component of the cost, is the additional computation cost since we have a larger non-private dataset which can increase the computation cost linearly. Moreover, Algorithm 2 uses an additional gradient computation call on the public data which will increase the computation cost. Training a model using our approach is about $2 - 3$ times slower than using the same model using DP-SGD without augmented images (without considering the time to generate the augmentations.).

## D    IMPERFECT PUBLIC DATA

One of the main downsides of the in-distribution public dataset is the cost of collecting such a dataset. As a result many applications are starting to use other public datasets which are not exactly from the same data distribution for differentially private deep neural networks. We call such data sets imperfect data distributions which can be from a slight different distribution or have noise data or noisy labels. While many public data algorithms require the data to be in-distribution, the existing works showed that it is still helpful to pretrain on the out-of-distribution public datasets Amid et al. (2021); Li et al. (2022a). In many cases naively using the imperfect public dataset can negatively affect the utility of the learning performance.

This is because the main assumption in Algorithm 2 is that the gradient on the public dataset is a very close estimation of the private data. However, this is not true for the out-of-distribution dataset. In order to fix this assumption, we rewrite Algorithm 2 by modifying line 9 to the following two lines which allow us to limit the effect of the public data:

$$\nabla_\theta^{s'} = \nabla_\theta^s \times \frac{\lambda}{\max(\lambda, ||\nabla_\theta^s||_2)}$$
$$\widetilde{\nabla_\theta^{(x,y)}} = \nabla_\theta^{s'} + ((\nabla_\theta^{(x,y)} - \nabla_\theta^{s'}) \times \frac{C}{\max(C, ||\nabla_\theta^{(x,y)} - \nabla_\theta^{s'}||_2)}) \tag{2}$$

To show the effect of imperfect data in the training, we evaluate the effect of other datasets from different but similar distributions as public data (different distribution) and noisy public dataset. We used the CIFAR100 dataset as the public data dataset for CIFAR10, and WikiText-3 for WikiText-2. While many of the previous works showed that pretraining on the dataset from a slightly different distribution (such as CIFAR100 for CIFAR10) can help improve the accuracy of the original model, we also show our additional techniques can also boost the accuracy. In particular, we observe that if we include all of the training dataset in the private training it will reduce the overall test accuracy, however, by including a random subset of the public dataset in the private training part can improve the privacy-utility trade-off. In our experiments for CIFAR10 using $10\%$ of the CIFAR100 gave us

Table 11: An ablation study on the effect of different techniques using imperfect public data on CIFAR10 models trained under $(2, 10^{-5})$-DP. We used CIFAR100 as public data for the CIFAR10 dataset.

| Settings | Test Acc (%) |
|---|---|
| Baseline (WRN16-4 pretrained on CIFAR100) | 75.9% |
| + Including the 10% of public data in the training dataset (extended) | 76.4% |
| + Using DOPE-SGD (Algorithm 2) (warm) | 77.1% |
| + Using Ensemble models | 77.3% |

Table 12: Test accuracy/perplexity models trained with differential privacy with $\delta = 10^{-5}$ and imperfect data (CIFAR100 for CIFAR10 and WikiText-3 for WikiText-2).

| | CIFAR10 (Test Acc) | | WikiText-2 (Test Ppl) | |
|---|---|---|---|---|
| $\varepsilon$ | DPSGD (warm) | DOPE-SGD | DPSGD (warm) | DOPE-SGD |
| 1.0 | 68.9% | **76.3**% | 79 | 77 |
| 2.0 | 76.4% | **77.3**% | 78 | 68 |
| 4.0 | 79.2% | **81.5**% | 76 | 65 |
| 6.0 | 82.5% | **84.9**% | 75 | 62 |

the highest boost in accuracy. For WikiText-2 dataset since the public dataset (Wikitext-3) is much more similar to the private dataset we do not see any downside of including all of the public data in the private training.

In Table 11, we study the effect of each technique on when the public data does not come from the same distribution as the private dataset. As we can see from the results, similar to the in-distribution dataset, using DOPE-SGD improves the utility further. Finally, in Table 12, we show the results for different dataset and privacy budgets. As we see, in all settings our method outperforms current approaches.

## E  ENSEMBLE OF PRIVATE MODELS WITHOUT ADDITIONAL DATA

As mentioned in Section 3.2, we can also apply the idea of ensemble of the private models to improve the utility of the final model without additional privacy cost. In the main body of this work, we show the effectiveness of this approach when we have public data. However, this approach can also be used in cases when we do not have public data. In Figure 4, we compare the accuracy of different ensemble approaches. As we see using majority voting can achieve higher accuracy compared to the other approaches. We also evaluate the effect of the number of models we use in the ensemble in Figure 5. In our experiments we observed that we need at least 50 models in an ensemble to have a noticeable gap between the final model and the ensemble model.

## F  PROOFS

*Proof of Proposition 2.* We start by analyzing the sensitivity of the gradient update rule. The update before adding noise and public gradient is equal to

$\nabla[t] = |B_t| \cdot \hat{g} + \sum_{(x,y) \in B_t} \frac{(\nabla^{(x,y)} - \hat{g}) \times C}{\max(C, \|\nabla^{(x,y)} - \hat{g})\|_2)}$. Note that $\hat{g}$ is data independent and comes from public data, therefore we only need to understand the sensitivity of the sum of clipped gradients. Since each example $(x,y)$ only affects one of the clipped gradients $\frac{(\nabla^{(x,y)} - \hat{g}) \times C}{\max(C, \|\nabla^{(x,y)} - \hat{g})\|_2)}$ and each of these vectors have a norm bounded by $C$, therefore the sensitivity of the sum is $C$.

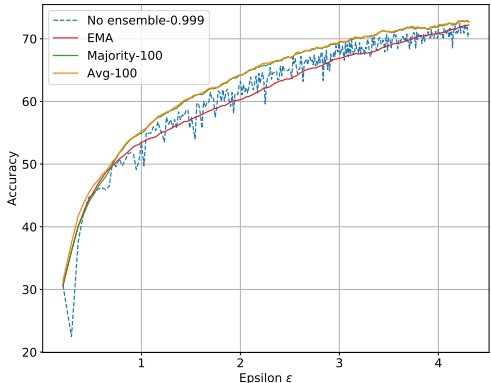

Figure 4: Comparison of the ensembling the last 100 models, the exponential moving average (EMA with decay rate 0.999) and the model without using any ensembling techniques for DP-SGD and any additional data on test accuracy of CIFAR10.

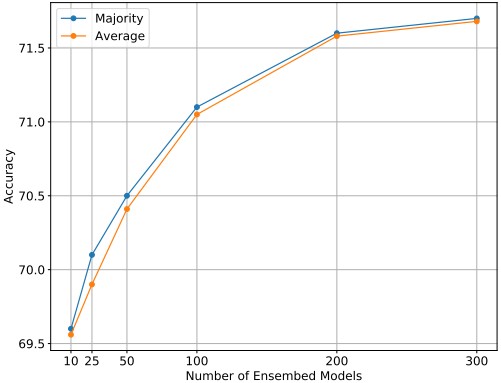

Figure 5: Comparison of the number of models in the ensemble on the accuracy for WRN-4-16 on CIFAR10 dataset with $\varepsilon = 4$, in our implementation of De et al. (2022) we observed a slight difference in the accuracy and the reported accuracy in their work. De et al. (2022) reported 71.4% for CIFAR10 dataset with $\varepsilon = 4$ but we were only able to achieve 71.1%.

Now, since we are adding Gaussian noise, each iteration is an instantiation of the sub-sampled Gaussian mechanism with sensitivity $C$, sampling rate $\frac{|B_t|}{|D|}$ and noise $\sigma$. Therefore, all the existing analysis for general DP-SGD (without additional assumptions) would also apply to Algorithm 1. $\square$

*Proof of Proposition 4.* As stated in proof of Proposition 2, each iteration of Algorithm 1 is a Gaussian mechanism with sampling rate $\frac{|B_t|}{|D|}$ and noise multiplier $\sigma/r$. Therefore, each iteration will be $(\frac{r \cdot \sqrt{2\ln(1.25/\delta')}}{\sigma}, \delta')$-DP. Assuming $\varepsilon' = \frac{r \cdot \sqrt{2\ln(1.25/\delta')}}{\sigma} < 1$, the sub-sampled mechanism will be $(2q \cdot \varepsilon', q\delta')$-DP. Then, by using advanced composition theorem for DP, we have that the composition of all $T$ steps is $(4Tq^2 \cdot \varepsilon'^2 + 2q \cdot \varepsilon' \cdot \sqrt{2T \cdot \log(1/\hat{\delta})}, Tq\delta' + \hat{\delta})$-DP. Assuming $\varepsilon' < \sqrt{\frac{\ln(1/\hat{\delta})}{2q^2 T}}$, the composition of $T$ mechanisms is $(4q\varepsilon'\sqrt{2T\ln(1/\hat{\delta})}, T\delta' + \hat{\delta})$-DP. Now setting $\hat{\delta} = \delta/2$ and $\delta' = \delta/2n$, the entire mechanism is $(4\frac{qr}{\sigma}\sqrt{2T\ln(2.5 \cdot T/\delta)\ln(2/\delta)}, \delta)$-DP which in turn implies $(4\sqrt{10}\frac{qr}{\sigma}\sqrt{T\ln(T/\delta)\ln(1/\delta)}, \delta)$-DP. In order for the assumptions to be correct, we need $\sigma > \max(2r \cdot \sqrt{\frac{q^2 T \ln(2.5 \cdot T/\delta)}{\ln(2/\delta)}}, r \cdot \sqrt{2\ln(2.5 \cdot T/\delta)})$. The privacy analysis for DP-SGD follows similarly, the only thing that changes is that we should use the clipping threshold $\mathcal{L}$ instead of $r$.

We would now prove that the output of Algorithm 1 is same as DP-SGD. Note that since the loss function is $\mathcal{L}$-lipschitz, the clipping operation for DP-SGD is a non-operational. Similarly, for DP-SGDA, since we have $r$-concentrated gradients, the clipping operation is non-operational. Given that the clipping operation is the only difference between DP-SGD and DP-SGDA, and they both are non-operational, the output distributions of the two algorithms are exactly the same. $\square$

