# OpenReview forum: "Effectively using  public data in privacy preserving Machine learning"
_ICLR.cc/2023/Conference — Submitted to ICLR 2023_

### Official Review · Reviewer_uKD4 · 2022-10-23

**Confidence:** 3
**Correctness:** 2
**Technical Novelty And Significance:** 3
**Empirical Novelty And Significance:** 2
**Recommendation:** 5

**Clarity, Quality, Novelty And Reproducibility:**

#Quality

The paper describes, justifies, implements and evaluates idea 3 thoroughly. Research questions are well chosen. The contribution is relevant to current research and is useful to practitioners.

#Clarity

The paper is unclear and does not explain experimental settings well at all. Here are questions that it leaves unanswered.
- does setting "warm" table 1 undo the augmentation applied in warm-aug ?
- which hyperparameters are fine-tuned in each experiment (cf. claim in section Settings)
- the text seems to use "generated data", "synthesized data", "augmentation" interchangeably in various places, which is confusing, but doesn't say this explicitly -- is this correct?
- does setting "extended" use augmentation or not?
- which ensembling technique is applied in table 1 experiments?
- how does majority voting apply to language modelling? It makes sense in classification, but in language modelling do you apply renormalization after probability averaging?
- does table 2 apply ensembling for LM ? It seems to apply it to CIFAR-10 since I see 75.1% like in table 1.
- what does Fed-SGD refer to exactly? maybe cite the work which is followed
- table 3, are "methods" respectively Amid+ 2021 and Li+ 2022?
- table 3, does Augmentation cover "warm-aug" or anything else too? I would recommend using a codename for settings, like warm-aug, defining it in one place, and then avoiding the use of periphrases to refer to it.
- where is out of distribution data used at all? why is it defined in section 4?
- in the Abstract, where does the number 68% come from? Is it the 68.1% in table 2 ?

#Originality

Using the numbering from my summary, ideas 3 and 4 seem novel, especially idea 3. It is most similar to Li+ 2022 where gradients are re-scaled instead of re-centered. Idea 1 is the standard way of using public data, idea 2 (data augmentation) seems straightforward, I find it hard to qualify it as novel. Idea 4 is surely a good idea that deserves being applied systematically, but is known to apply quite generally.


#Typos and errors

- Table 5, column eps=2, 40k: the figure should be 75.1% I think?
- conclusion, first sentence is ungrammatical
- "why this method works" is vague
- fig 2 caption: "the noise added by the Gaussian mechanism": do you refer to the Gaussian noise term used in DP-SGD? clarify
- As we increase the privacy budget...: sentence very imprecise (my suggestion: use commas more often)
- section Comparison: first 6 lines are incoherent

**Strength And Weaknesses:**

Strengths:
- convincing demonstration of superiority
- good set of experiments

Weaknesses:
- exposition unclear, verging on obscure
- experimental settings insufficiently documented


**Summary Of The Paper:**

This paper uses several ideas to improve on the privacy-utility trade-off of DP-SGD using public data, beyond the traditional approach of pre-training on public data:
1) use data augmentation to pre-train on augmented data (setting called "warm" in paper)
2) include public data in training data (setting "warm-aug")
3) at each DP-SGD gradient calculation step, before clipping and noise addition, recenter the gradient vector using a gradient estimate learnt from public data (technique called DOPE-SGD in paper)
4) ensembling (I seem to understand that means "using majority voting over n checkpoints" on CIFAR-10 experiments)


Experiments mainly on CIFAR-10, but also on language modelling are shown. Ablation and analytical experiments are conducted. The combination of methods improves on the state of the art.

**Summary Of The Review:**

The paper's central claim of beating SOTA seems supported by experimental evidence on both CIFAR-10 and language modelling, in different settings. The description of experimental settings and protocols is unclear and leaves much underspecified. I see no major flaw in the work and it has potential to be improved.

---

> ### Author Response · Authors · 2022-11-16
> **Responses to reviewer uKD4**
>
> #Clarity:
> We want to thank the reviewer for detailed comments. We fixed all of the issues and also we have the responses here:
>
> "> does setting "warm" … ": No, this was a typo we fixed in the paper.
>
> "> which hyperparameters  …":  We added the reference to the implementation detail section in Appendix A, to answer the question we fine tuned: learning rate, noise multiplier, size of public data, clipping norm, batch size.
>
> "> the text seems to use … ": We edited the paper to make the wording more consistent.
>
> "> does setting "extended" … ": Yes, fixed in the paper.
>
> "> which ensembling technique is applied in table 1 experiments?": We used the maximum between two techniques in all experiments, In particular for table 1 we used majority voting.
>
> "> how does majority voting apply to language modeling": We take the average of the probabilities between the multiple models, we added the details to the paper
>
> "> what does Fed-SGD ..." : Citation added, fed-sgd updates the model parameters after each batch.
>
> "> table 3, are "methods" : Yes, Added citations for clarity.
>
> "> table 3, does Augmentation...": We added the details in the appendix to clarify all of the settings we used in the paper.
>
> "> where is out of distribution data...": We added reference to the out-of-distribution experiments which are in Appendix D.
>
> "> in the Abstract, where ...": Yes it is coming from table 2, replaced with the exact value.
>
> #Originality
>
> We are glad that the reviewer found some of our ideas novel. For idea 1, we didn’t find any existing works that studies using diffusion models as augmentation technique for pretraining. While the idea of pretraining is not new, training a diffusion model on the public data and using this for pre-training has not been studied before. Similarly we did not find any work about including the public data in the private training part.
>
> #Typos
>
> Fixed all of them in the paper.
>
> #Correctness
>
> The reviewer also mentioned that some claims in the paper are not well supported or correct, we would be happy to add additional details if the reviewer specified which claim they are referring to.

---

### Official Review · Reviewer_Y2rR · 2022-10-25

**Confidence:** 4
**Correctness:** 3
**Technical Novelty And Significance:** 3
**Empirical Novelty And Significance:** 2
**Recommendation:** 6

**Clarity, Quality, Novelty And Reproducibility:**

Overall the paper is on the borderline of acceptance and rejection. I am happy to discuss more and to consider raise my score if my concerns are addressed.

**Strength And Weaknesses:**

Strength: This paper is well-written and the methodology is comprehensive, including different aspects of using public data (or public gradient) in the DP training. The experiments are carefully conducted and I appreciate the ablation study in Table 1. I think the new algorithms are interesting with good motivation.

Weaknesses: 1. I think Algorithm 1&2 fall in the category of weighted sum between public and private gradients. Let's denote $g_i$ as per-sample grad, $\hat g$ as public gradient like in Line 7 of Algorithm 1. Then the clipped residual is $c_i*(g_i-\hat g)$ and Line 9 becomes $(1-c_i)\hat g+c_i g_i$. Then the summed clipped gradient is $\sum_i (1-c_i)\hat g+\sum_i c_i g_i$ and the noise is added. Clearly, this is a weighted sum that has been also studied in https://arxiv.org/abs/2111.00115 and several other works. So I think they should be discussed. 2. I wonder do the authors experiment with the average angle between clipped per-sample gradients and the public gradient at each iteration? This could be very helpful to convince that your new clipping helps reduce the bias (also, only the angle bias not the magnitude matters, as the latter can be recovered with adjusting learning rate). 3. The experiments do not show significant improvement but the algorithms are seemingly over-complicated.

**Summary Of The Paper:**

This paper studies multiple ways to incorporate public data to help private training,  (1) use pre-training to improve the initial point of private training, (2) use a novel DP-SGD algorithm during training, (3) post process on the private models. The contribution is complete, tackling aspects before/during/after the training. Ablation studies are done on CIFAR10 and Wikitext to show the performance.

**Summary Of The Review:**

I think the paper is well-written but the experiments seem thin and the results aren't very exciting.

---

> ### Author Response · Authors · 2022-11-16
> **Responses to reviewer Y2rR**
>
> "> I think Algorithm 1 & 2 fall in the …":
>
> While the reviewer's formulation is correct, it is important to note that c_i is still a data dependent value and not a constant. In contrast, in the paper mentioned by the reviewer, the linear combination is data-independent. We do not see a way to instantiate the mechanism of the aforementioned paper to simulate our algorithm. We also note that the referred paper calculates a weighted average between public and private mean on non machine learning tasks. Moreover, mirror gradient descent paper [Amit et al] uses the idea of a weighted sum between the public and private gradient which we compared in our paper. Nonetheless, we will add this work as a related work.
>
> "> average angle between clipped per-sample gradients and the public gradient at each iteration?":
>
> We argue that public data should not be used as a reference for measuring the bias of privatized gradients. As training continues, the overfitting to the public data increases and the angle between public and private gradient becomes less informative.
>
> Instead in Figure 2, we measure the dot product between the privatized gradients and the actual  gradient of private data for dp-sgd and dope-sgd. We show that our approach has a higher dot product, which means the private gradients in dope-sgd are closer to the true gradient of the batch compared to dp-sgd which can produce less bias.
>
> Please note that while size of the gradient vectors can be modified by the learning rate, as can be seen in Figure 2 gradient sizes vary a lot among iterations and we cannot easily scale them up or down by changing the learning rate.
>
>
>
> "> The experiments do not show significant improvement …":
>
> If we compare the second line in table 2 to the last line of the table, we see about 7% improvement for epsilon 2 using all techniques in our work. Moreover, each of our techniques can individually improve the accuracy of the existing works significantly. The improvements are more significant in the simpler task and settings as well  (Table 9 and 10, appendix B).

---

### Official Review · Reviewer_d916 · 2022-10-25

**Confidence:** 2
**Clarity, Quality, Novelty And Reproducibility:** The paper is clearly presented with a…
**Correctness:** 3
**Technical Novelty And Significance:** 3
**Empirical Novelty And Significance:** 3
**Recommendation:** 6

**Strength And Weaknesses:**

Strength:
- The techniques are well-motivated and presented. The intuitions behind the proposed techniques are well explained. A solid theoretic backbone is also provided for their claims.
- Those techniques are novel and not very complicated. They also provide good experiments demonstrating the impacts and improvements in different settings and parameters.

Weakness:
- It's better to have proofs of proposition 2 and 4 in the appendix.

**Summary Of The Paper:**

This paper proposes new techniques for using public data in differentially private machine learning and makes a significant improvement over the state-of-the-art solution. More specifically, they propose to use synthesized data provided by a generative model trained by the given public data, and a new gradient clipping mechanism that achieves higher accuracy with the help of public and synthesized data.

**Summary Of The Review:**

Overall I found no flaw in this paper and would recommend it.

---

> ### Author Response · Authors · 2022-11-16
> **Responses to reviewer d916**
>
> We added the proofs of propositions 2 and 4  to the paper in Appendix F.

---

### Official Review · Reviewer_UAzg · 2022-10-25

**Confidence:** 4
**Clarity, Quality, Novelty And Reproducibility:** This paper is well written and the me…
**Correctness:** 4
**Technical Novelty And Significance:** 4
**Empirical Novelty And Significance:** 3
**Recommendation:** 6

**Strength And Weaknesses:**

**Strength**
1. This paper is written very clear and organized.
2. It clearly explains the intuition why the new proposed clipped gradient would be potentially better than clipping to the origin. The approach is overall well-motivated.

**Weaknesses**

The experiments are mostly well-designed. However, it might miss one fair baseline, which is training only with (augmented) public data, including the new generated synthetic data. This baseline can justify the improvement from the second phase (new clipping method) and third phase (model ensemble) in the approach.

Minor issue:
1. Notation of *L* is re-used as both loss function in Algorithm 1&2 and the Lipschitz factor.
2. "Out-of-distribution public data" is introduced in the dataset descriptions but is not mentioned in the later results in the main text. (I checked that it appears in the experiments in Appendix).

**Summary Of The Paper:**

This paper studies the DP model training with public data. They utilize the public data in three phases:  pretraining, better optimization process and a post-process ensemble of private models. The experimental results show the significant improvement of their approach compared with the SOTA.

**Summary Of The Review:**

The results of proposed approach are very promising. The missing baseline (see weaknesses) leads to a some uncertainty that how much the DOPE-SGD helps, which will influence the justification of significance. Thus I give the score of 6.

---

> ### Author Response · Authors · 2022-11-16
> **Responses to reviewer UAzg**
>
> "> baseline with training only on augmented data...": Training on only augmented data gets 69.4% accuracy on average for CIFAR10 and perplexity of 240  on the wikitext model . We added the results to the paper in Table 2.
>
> "> Notation of L...": Thank you for pointing out, we have changed accordingly and consistently: $l$ for loss function and $\mathcal{L}$ for the Lipschitz constant.
>
> "> out-of-distribution data...": We will add a summary of the result to the main text. We moved it to the appendix due to the page limit. We will use a better organization to include some of the results in the main body.

---

### Decision · Program_Chairs · 2023-01-20

**Decision:**

Reject

**Justification For Why Not Higher Score:**

The paper has several major weaknesses. The conceptual basis of this work is not compelling and the technical assumptions are problematic.

**Justification For Why Not Lower Score:**

N/A

**Metareview: Summary, Strengths And Weaknesses:**

The paper studies leveraging public data in improving the training performance of DP-SGD. The central idea is to use the public data to reduce the gradient clipping value. The authors propose to do so by using the public data to provide an estimate for the gradient in each iteration of DP-SGD, then recenter the private gradients around such an estimate before clipping. The paper also discusses other techniques for leveraging public data that were studied in previous works such as pre-training and model ensembles. The authors conduct several experiments to evaluate the performance of their proposed methods.


Strengths:

- Some of the experimental results look promising, which is an indication that there is an interesting phenomenon worthy of more formal investigation.

- The general idea the authors is proposing seems intuitive and is based on what has been observed in some practical scenarios of deep learning.


Weaknesses:

- The paper fails to make a compelling conceptual reasoning and a sound formal analysis for the proposed method, mainly due to the reliance on a very strong assumption of uniform gradient concentration with probability 1, which makes the conceptual basis of this work quite shaky.  I am referring here to the notion of concentrated losses (which is actually used to mean concentrated gradients). First, this can only hold with certain probability not with probability 1 as it is used in the paper. Second, even if one allows this notion to hold only with certain probability (over the randomness of the i.i.d. private and public data), to ensure a tight enough concentration around the estimated gradient from a mini-batch of the public data, the private gradient needs to be computed from a large enough batch of private samples, not from on a single private data point as it is done in the paper (see Algorithm 2).

- The main result asserting the equivalence of DP-SGD and their proposed algorithm completely relies on such an unrealistic assumption mentioned above.

- Several experimental results are not clearly discussed. The lack of transparency and clarity raises some doubts about how the experiments were conducted and their significance. The authors did not properly address those concerns in their rebuttal.

- The paper is not very well written. There are notational gaps that the readers have to fill in (e.g., the transition between Algorithm 1 and Algorithm 2). Some statements are also written in a cumbersome fashion, and there are various typos in the paper.

- Some recent works on adaptive gradient clipping with differential privacy are not discussed and compared to the proposed work, e.g., [A, B, C].

There is a consensus among the reviewers and the AC that there is a room for improvement in terms of the conceptual and formal reasoning, experimental work, and writing quality. In particular, the authors are encouraged to reconsider the main assumptions and think more about a compelling characterization of the phenomenon they observed in their experiments.

[A]: Andrew, G., Thakkar, O., McMahan, B., Ramaswamy, S. (2021). Differentially private learning with adaptive clipping. Advances in Neural Information Processing Systems, 34, 17455-17466.

[B]: Varshney, P., Thakurta, A., Jain, P. (2022). (Nearly) Optimal Private Linear Regression via Adaptive Clipping. arXiv preprint arXiv:2207.04686.

[C]: Chen, X., Wu, S. Z., Hong, M. (2020). Understanding gradient clipping in private SGD: A geometric perspective. Advances in Neural Information Processing Systems, 33, 13773-13782.

**Summary Of Ac-Reviewer Meeting:**

The AC and reviewers discussed the paper and converged to the following summary in terms of the strengths and weaknesses of this work.

Strengths:

- Some of the experimental results look promising, which is an indication that there is an interesting phenomenon worthy of more formal investigation.

- The general idea the authors is proposing seems intuitive and is based on what has been observed in some practical scenarios of deep learning.

Weaknesses:

- The paper fails to make a compelling conceptual reasoning and a sound formal analysis for the proposed method, mainly due to the reliance on a very strong assumption of uniform gradient concentration with probability 1, which makes the conceptual basis of this work quite shaky.

- The main result asserting the equivalence of DP-SGD and their proposed algorithm (Proposition 4) completely relies on such an unrealistic assumption mentioned above.

- Several experimental results are not clearly discussed. The lack of transparency and clarity raises some doubts about how the experiments were conducted and their significance. The authors did not properly address those concerns in their rebuttal.

- The paper is not very well written. There are notational gaps that the readers have to fill in (e.g., the transition between Algorithm 1 and Algorithm 2). Some statements are also written in a cumbersome fashion, and there are various typos in the paper.

- Some recent works on adaptive gradient clipping with differential privacy are not discussed and compared to the proposed work, e.g., [A, B, C].

The AC and reviewers agreed that there is a room for improvement in terms of the conceptual and formal reasoning, experimental work, and writing quality. In particular, the authors are encouraged to reconsider the main assumptions and think more about a compelling characterization of the phenomenon they observed in their experiments.

[A]: Andrew, G., Thakkar, O., McMahan, B., Ramaswamy, S. (2021). Differentially private learning with adaptive clipping. Advances in Neural Information Processing Systems, 34, 17455-17466.

[B]: Varshney, P., Thakurta, A., Jain, P. (2022). (Nearly) Optimal Private Linear Regression via Adaptive Clipping. arXiv preprint arXiv:2207.04686.

[C]: Chen, X., Wu, S. Z., Hong, M. (2020). Understanding gradient clipping in private SGD: A geometric perspective. Advances in Neural Information Processing Systems, 33, 13773-13782.